

# Multicriteria scheduling of two-subassembly products with batch availability and precedence constraints

Zhenxin Wen and Shuguang Li

School of Computer Science and Technology, Shandong Technology and Business University, Yantai, China

## ABSTRACT

This article studies the multicriteria problems of scheduling a set of n products on a fabrication facility, focusing on batch availability and precedence constraints. Each product is composed of two distinct subassemblies: a common subassembly, shared across all products, and a unique subassembly unique to each product. The common subassemblies are processed together in batches, with each batch requiring an initial setup, while unique subassemblies are handled individually. The availability of a common subassembly is contingent upon the completion of its entire batch (*i.e.*, batch availability), whereas a unique subassembly becomes available immediately after its processing. The product completion time is determined by the availability of both subassemblies. Strict (weak) precedence means that if a product precedes another, then the latter can start only after the former is completed (the latter cannot start earlier than the former). We propose O(n⁴)-time algorithms to simultaneously optimize makespan and maximum cost, as well as to lexicographically optimize two maximum costs and makespan under strict or weak precedence constraints.

# INTRODUCTION

Driven by the need to balance conflicting objectives, the field of multicriteria scheduling has garnered significant attention over the past few decades, see *Hoogeveen, 2005*, *T'kindt & Billaut (2006)*. This article studies four specific multicriteria problems related to scheduling products with two subassemblies on a fabrication facility, subject to either strict or weak precedence constraints. The focus is on optimizing both makespan and maximum cost simultaneously, or alternatively, on the lexicographical optimization of two maximum costs along with makespan.

In formal terms, consider a set $\mathcal{T}$ of $n$ products, denoted by $T_1, T_2, \ldots, T_n$, to be processed on a single fabrication facility. Each product, $T_j \in \mathcal{T}$, consists of two parts: a *common subassembly* $T_j^{(1)}$ and a *unique subassembly* $T_j^{(2)}$, with respective processing times $t_j^{(1)}$ and $t_j^{(2)}$. For aerospace component production, strict precedence constraints might arise when assembling engines: a turbine blade subassembly ($T_i$) must be fully completed before a combustion chamber subassembly ($T_j$) can start, ensuring safety and structural

Corresponding author
Shuguang Li, sgliytu@hotmail.com

integrity. The facility processes the common subassemblies in batches, where each batch incurs a *setup time $\delta$*, while the unique subassemblies are processed individually. The setup time for each unique subassembly is incorporated into its total processing time, as it is unique to the product it belongs to. Consequently, it is assumed that unique subassemblies do not require setup times. The two subassemblies of a product may be processed in either order, but the facility can handle only one subassembly at a time, and no preemption is allowed during processing.

In this article, we adopt the assumption of *batch availability* for the common subassemblies, meaning that a common subassembly becomes available only after the entire batch to which it belongs has been fully processed (*Santos & Magazine, 1985*) (Alternatively, there is the *item availability* assumption, where a common subassembly becomes available immediately after its processing is finished). Conversely, unique subassemblies are considered available as soon as their individual processing is completed. A product is deemed complete only when both its common and unique subassemblies are fully processed and available. Moreover, each product, $T_j \in \mathscr{T}$, is associated with two cost functions $f_j$ and $g_j$, which represent the costs incurred based on the product's completion time. It is assumed that these cost functions are *regular*, meaning $f_j$ and $g_j$ are non-decreasing with respect to the product completion times.

In practical terms, the cost functions $f_j$ and $g_j$ can be linked to real-world manufacturing metrics. $f_j(C_j)$ often represents lateness penalties, such as contractual fines for delivering products after their due dates ($d_j$). In just-in-time (JIT) production systems, this could model penalties for delaying shipments to assembly lines. $g_j(C_j)$ may quantify resource utilization costs, such as idle machine fees or inventory holding charges for early-completed products. In aerospace manufacturing, this might reflect the cost of storing specialized components before final integration. These costs are "regular" because they increase with completion time, a common assumption in scheduling problems to align with real-world inefficiencies.

For a given schedule $S$, let $C_j(S)$ represent the completion time of product $T_j$ in $S$. Define $f_j(C_j(S))$ and $g_j(C_j(S))$ as two costs associated with $T_j$ in the schedule. The values $f_{\max}(S) = \max_j f_j(C_j(S))$ and $g_{\max}(S) = \max_j g_j(C_j(S))$ correspond to the maximum costs under these criteria. Notably, two specific cases of maximum cost are the makespan, $C_{\max}(S) = \max_j \{C_j(S)\}$, and the maximum lateness, $L_{\max}(S) = \max_j \{C_j(S) - d_j\}$, where $d_j$ denotes the due date for product $T_j$. The argument $S$ can be omitted in the notation whenever the context is clear.

In addition, the problems under consideration have either strict or weak precedence constraints. A product may depend on a set of products that must be completed or started before it can start. Formally speaking, for the *strict precedence relation* $\prec$, if $T_i \prec T_j$ ($T_i$ precedes $T_j$ because of a higher priority), then $T_j$ can start only after $T_i$ is completed in any feasible schedule. Consequently, the common subassemblies of $T_i$ and $T_j$ must be in different batches. As for the *weak precedence relation* $\preceq$, if $T_i \preceq T_j$, then $T_i$ must start no later than $T_j$ in any feasible schedule. Therefore, the common subassemblies of $T_i$ and $T_j$ may be processed within the same batch.

The first two problems examined in this article focus on identifying Pareto optimal schedules that minimize both the makespan $C_{max}$ and the maximum cost $f_{max}$ simultaneously, while adhering to either strict or weak precedence constraints. Utilizing the notation conventions established in *Hoogeveen (2005)*, *T'kindt & Billaut (2006)*, *Brucker (2007)*, these problems are represented as $1| \prec, 2 - subpt, BA|(C_{max}, f_{max})$ and $1| \preceq, 2 - subpt, BA|(C_{max}, f_{max})$, where "$\prec$" and "$\preceq$" indicate strict and weak precedence constraints, respectively. The symbol "$2 - subpt$" refers to "two-subassembly products", and "$BA$" denotes "batch availability".

Let $\rho$ and $\tau$ represent the two performance criteria to be minimized. A schedule $S$ is considered Pareto optimal or non-dominated if there exists no other feasible schedule $S'$ for which $\rho(S') \leq \rho(S)$ and $\tau(S') \leq \tau(S)$ where at least one of these inequalities is strict. When a schedule $S$ meets these conditions, the resulting objective vector $(\rho(S), \tau(S))$ is termed a Pareto optimal point (*Hoogeveen, 2005*). This approach is referred to as Pareto optimization or simultaneous optimization.

The last two problems addressed in this article involve finding a lexicographically optimal schedule under strict or weak precedence constraints, where the criteria $f_{max}$, $g_{max}$, and $C_{max}$ are prioritized as primary, secondary, and tertiary objectives, respectively. These problems are represented as $1| \prec, 2 - subpt, BA|Lex(f_{max}, g_{max}, C_{max})$ and $1| \preceq, 2 - subpt, BA|Lex(f_{max}, g_{max}, C_{max})$. An optimal solution for $1| \cdot |Lex(\rho_1, \rho_2, \ldots, \rho_k)$ is defined as the best possible schedule for $1| \cdot |\rho_k$ within the subset of schedules that are already optimal for $1| \cdot |Lex(\rho_1, \rho_2, \ldots, \rho_{k-1})$. This approach is known as lexicographical optimization or hierarchical optimization (*Hoogeveen, 2005*).

The structure of this article is as follows: "Literature Review" offers a review of relevant research in the field. "Algorithms for Pareto Scheduling Two-subassembly Products with Precedence Constraints" introduces $O(n^4)$-time algorithms for $1| \prec, 2 - subpt, BA|(C_{max}, f_{max})$ and $1| \preceq, 2 - subpt, BA|(C_{max}, f_{max})$. "The Experiments", the experiment of the Algorithm $C_{max}$-$F_{max}$ is given and compared with the algorithm of $1|2 - subpt, BA|(C_{max}, f_{max})$. "Algorithms for Lexicographical Scheduling Two-subassembly Products with Precedence Constraints" presents $O(n^4)$-time algorithms for $1| \prec, 2 - subpt, BA|Lex(f_{max}, g_{max}, C_{max})$ and $1| \preceq, 2 - subpt, BA|Lex(f_{max}, g_{max}, C_{max})$. In conclusion, "Conclusions" discusses potential directions for future research.

## LITERATURE REVIEW

For an in-depth exploration of multicriteria scheduling and batch scheduling, readers are encouraged to consult the surveys provided in *Hoogeveen (2005)*, *T'kindt & Billaut (2006)*, *Herzel, Ruzika & Thielen (2021)* and *Potts & Kovalyov (2000)*, *Allahverdi et al. (2008)*, respectively. In this article, we focus specifically on discussing results that are directly relevant to the scheduling of two-subassembly products within the context of batch availability.

*Baker (1988)* initially introduced the model of scheduling two-subassembly products with the goal of minimizing total completion time, denoted as $1|2 - subpt, BA|\Sigma C_j$. Operating under the agreeability assumption—where the processing time of a product's

common subassembly is shorter than another's whenever the same relationship holds for their unique subassemblies—he devised an $O(n^2)$-time algorithm. Later, *Coffman et al. (1990)* enhanced this algorithm, reducing the runtime to $O(n \log n)$. Subsequently, *Gerodimos, Glass & Potts (2000)* proposed an $O(n^2)$-time dynamic programming algorithm for $1|2 - subpt, BA|L_{\max}$. They also demonstrated that the problem of minimizing the total number of late products, denoted as $1|2 - subpt, BA|\Sigma U_j$, is NP-hard (here, $U_j = 1$ if $C_j > d_j$ and $U_j = 0$ otherwise). To address $1|2 - subpt, BA|\Sigma U_j$, they provided a pseudo-polynomial time dynamic programming solution. Furthermore, for the special case of $1|2 - subpt, BA|\Sigma U_j$ where all common subassemblies share identical processing times, they developed an $O(n^4 \log n)$-time dynamic programming algorithm. *Wagelmans & Gerodimos (2000)* later refined the algorithm in *Gerodimos, Glass & Potts (2000)* for $1|2 - subpt, BA|L_{\max}$, reducing the runtime to $O(n \log n)$. *Yang (2004a)* explored a different scenario where common subassemblies are divided into multiple families, each with setup times that are sequence-independent but not identical. He introduced a branch-and-bound algorithm aimed at minimizing total completion time. In a separate study, *Yang (2004b)* investigated the problem in the context of parallel machines, with the constraint that both subassemblies of a product must be processed on the same machine. He proposed two heuristic approaches to generate near-optimal schedules for minimizing total completion time. *Li (2023)* examined $1|2 - subpt, BA|(C_{\max}, L_{\max})$, a bicriteria scheduling problem for two-subassembly products on a single facility, focusing on minimizing both makespan and maximum lateness, and developed an $O(n^2 \log n)$-time algorithm with linear memory usage.

If all products consist solely of common subassemblies (*i.e.*, all $t_j^{(2)} = 0$), then the problem of scheduling two-subassembly products simplifies to the serial-batch scheduling problem. This problem can be modeled in two ways based on the batch capacity—the largest number of products that can be processed within a single batch. The first is the *bounded model*, where the batch capacity is limited, denoted as $b < n$, and the second is the *unbounded model*, where the batch capacity is unlimited, denoted as $b \geq n$.

Over the past two decades, serial-batch scheduling problems have received extensive attention and investigation (*Baptiste, 2000*; *Cheng & Kovalyov, 2001*; *Yuan, Yang & Cheng, 2004*; *Ng, Cheng & Yuan, 2002*; *Webster & Baker, 1995*; *He, Lin & Yuan, 2008*; *He et al., 2013a, 2013b*; *He, Lin & Lin, 2015*; *Geng, Yuan & Yuan, 2018*). Among the contributions, *Baptiste (2000)* introduced an $O(n^{14} \log n)$-time algorithm for $1|SUB, r_j, p_j = p|L_{\max}$ and $1|SBB, r_j, p_j = p|L_{\max}$, where products have varying release dates but identical processing times. In this context, "*SUB*" refers to the unbounded model of serial-batch scheduling, while "*SBB*" refers to the bounded model. *Cheng & Kovalyov (2001)* explored serial batch scheduling problems with the objective of minimizing various regular cost functions. For products with equal release dates, they developed dynamic programming algorithms aimed at minimizing several key criteria: maximum lateness, the number of late products, total tardiness, total weighted completion time, and total weighted tardiness, under the condition of equal due dates. These algorithms are polynomial when the number of distinct due dates or processing times is fixed. Additionally, they proposed more efficient

algorithms for certain special cases and established the NP-hardness for several specific cases of the bounded model.

By suitably adjusting the release dates and due dates, problems $1| \preceq, SUB, r_j, p_j = p|L_{\max}$ (involving weak precedence constraints) and $1| \preceq, SUB|L_{\max}$ (where products have varying processing times but identical release dates) can be transformed in $O(n^2)$ time into $1|SUB, r_j, p_j = p|L_{\max}$ and $1|SUB|L_{\max}$ (*Yuan, Yang & Cheng, 2004*; *Ng, Cheng & Yuan, 2002*). These transformed problems can then be solved in $O(n^{14} \log n)$ and $O(n^2)$ time, respectively (*Baptiste, 2000*; *Webster & Baker, 1995*).

*He, Lin & Yuan (2008)* developed an $O(n^2)$-time algorithm for problem $1|SUB|(C_{\max}, L_{\max})$. This result was subsequently enhanced to an $O(n^5)$-time algorithm (*He et al., 2013a*), and further improved to an $O(n^3)$-time algorithm for problem $1|SUB|(C_{\max}, f_{\max})$ (*He et al., 2013b*). *He, Lin & Lin (2015)* also proposed an $O(n^6)$-time algorithm for $1|SBB|(C_{\max}, L_{\max})$. Additionally, *Geng, Yuan & Yuan (2018)* introduced $O(n^4)$-time algorithms for both $1|SBB|(C_{\max}, f_{\max})$ and $1| \prec, SUB|(C_{\max}, f_{\max})$. They also devised an $O(n^2)$-time algorithm for problem $1| \preceq, SUB|(C_{\max}, L_{\max})$, and demonstrated that problems $1| \preceq, SBB, b = 2|L_{\max}$ and $1| \prec, SBB, b = 2|L_{\max}$ are strongly NP-hard.

This article addresses multicriteria scheduling problems with theoretical significance: optimizing makespan and maximum cost under batch availability and precedence constraints. The polynomial-time complexity of our proposed $O(n^4)$ algorithms is critical: it demonstrates it is computationally feasible to obtain optimal solutions for these problems, contrasting with NP-hard scheduling problems where only heuristic solutions exist.

To the best of our knowledge, problems $1| \prec, 2 - subpt, BA|(C_{\max}, f_{\max})$, $1| \preceq, 2 - subpt, BA|(C_{\max}, f_{\max})$, $1| \prec, 2 - subpt, BA|Lex(f_{\max}, g_{\max}, C_{\max})$ and $1| \preceq, 2 - subpt, BA|Lex(f_{\max}, g_{\max}, C_{\max})$ have not been studied in previous research. In this article, we present $O(n^4)$-time algorithms to solve each of these problems. Note that (*Li, 2023*) focused on a specific variant of the first problem, where products are not subject to precedence constraints, and the objective is to minimize maximum lateness rather than maximum cost.

While earlier studies focused on single-objective batch scheduling, recent research has shifted toward multicriteria optimization in dynamic manufacturing systems. *Li (2024)* proposed a hybrid heuristic for two-subassembly scheduling with energy consumption constraints, extending the batch availability model to sustainable manufacturing. *Hidri & Tlija (2024)* addressed sequence-dependent setup times in hybrid flow shops, a scenario analogous to our unique subassembly processing with individual setup costs. These advancements highlight the growing need to model real-world complexities like dynamic precedence rules and multi-echelon inventory constraints.

Additionally, *Li (2024)* examined the bounded model for scheduling two-subassembly products with equal processing times for all common subassemblies. *Hidri & Tlija (2024)* and *Xu et al. (2024)* introduces a heuristic algorithm to solve this complex problem. The flexible job shop scheduling problem (FJSP) is (*Serna et al., 2021*) combination problem. In this context, *Li (2024)* developed an $O(n^2 \log n)$-time algorithm for the simultaneous

optimization of makespan and maximum lateness, along with an $O(n^4)$-time algorithm for the lexicographical optimization involving two maximum lateness objectives and makespan.

# ALGORITHMS FOR PARETO SCHEDULING TWO-SUBASSEMBLY PRODUCTS WITH PRECEDENCE CONSTRAINTS

In this section, we will present an $O(n^4)$-time algorithm designed to solve problem $1| \prec, 2 - subpt, BA|(C_{\max}, f_{\max})$. Additionally, the final schedule produced by this algorithm is also optimal for the single criterion problem $1| \prec, 2 - subpt, BA|f_{\max}$. Towards the end of this section, we will demonstrate how minor adjustments to the algorithm allow it to efficiently solve $1| \preceq, 2 - subpt, BA|(C_{\max}, f_{\max})$ in $O(n^4)$ time as well.

Precedence constraints on $\mathcal{T}$ can be represented using a graph $G = <V, E>$, where each vertex in $V$ corresponds to a product in $\mathcal{T}$, and each edge in $E$ represents a pair $<T_p, T_j>$ indicating that $T_p \prec T_j$, i.e., $T_p$ must precede $T_j$.

In the context of automotive component manufacturing, a workshop's production scheduling problem for gearbox systems can be abstracted into the dual-component product scheduling model studied in this article. Specifically, each gearbox product consists of two types of subassemblies: one is a common subassembly shared by all models (e.g., the gearbox housing), which must be processed in batches with a 2-h mold setup time ($\delta = 2$) required before each batch production; the other is a customized unique subassembly (e.g., the gear set) for different vehicle models, which requires no additional preparation time and can be processed individually. During production, strict assembly sequence constraints exist between some models—for example, the basic-type gearbox must be completed before the high-performance type—corresponding to the strict precedence constraint ($T_i \prec T_j$) in the model, meaning that the successor product can only start production after both the common and unique subassemblies of the predecessor product are completed. This scenario necessitates simultaneous optimization of two key objectives: minimizing makespan ($C_{\max}$) to enhance equipment utilization by reducing idle time of the CNC machining center, and minimizing maximum delay cost ($f_{\max}$) to avoid contractual penalty fees for late deliveries. This scenario necessitates simultaneous optimization of two key objectives: minimizing makespan ($C_{\max}$) to enhance equipment utilization by reducing idle time of the CNC machining center, and minimizing maximum delay cost ($f_{\max}$) to avoid contractual penalty fees for late deliveries.

The integration of these constraints and objectives is visualized in the following production scheduling flowchart (Fig. 1), which illustrates the batch processing of common subassemblies, individual machining of unique subassemblies, and the enforcement of strict precedence relationships within the dual-component scheduling framework.

Recall that Lawler's algorithm (*Lawler, 1973*) solves $1| \prec |f_{\max}$ (the problem of minimizing maximum cost under precedence constraints on a single facility) in $O(n^2)$ time. The algorithm constructs the schedule in reverse order, relying on the following key

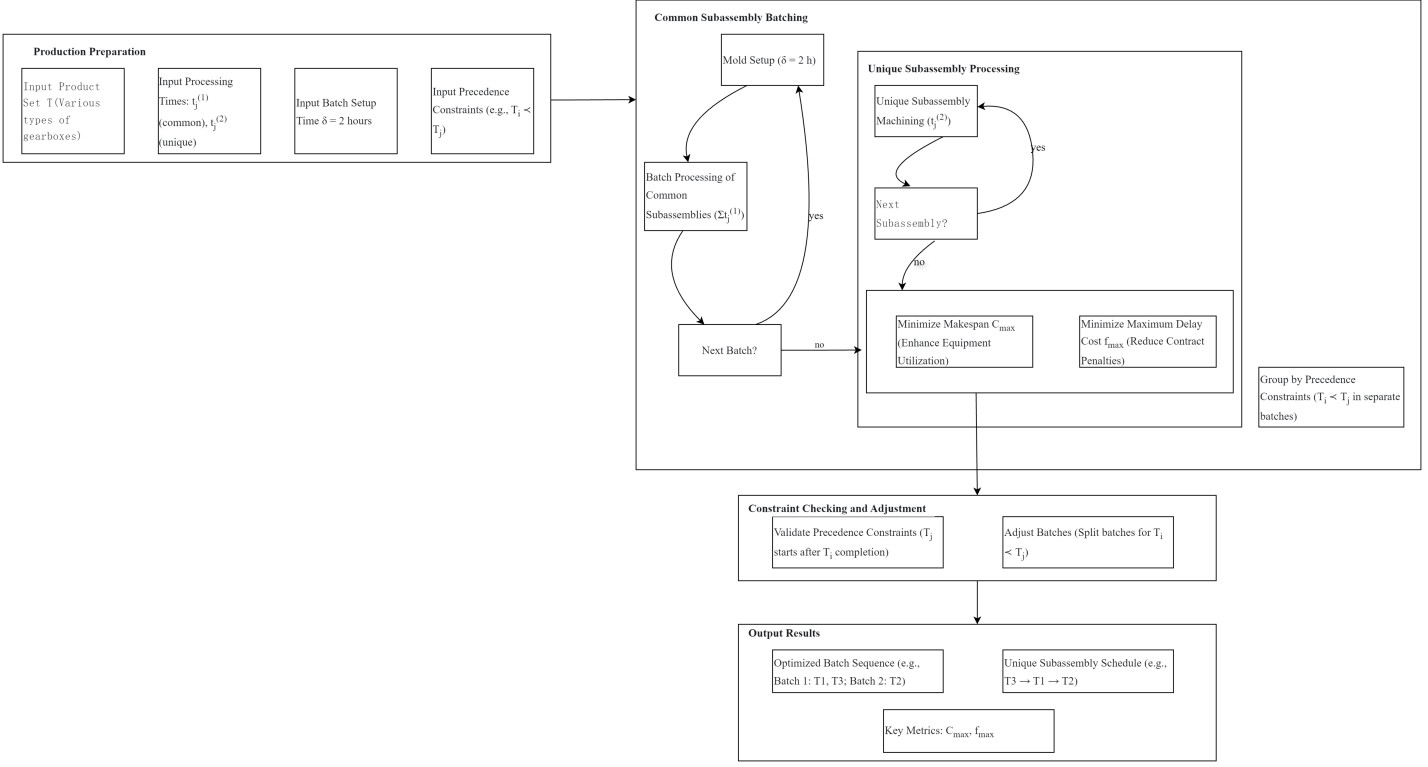

**Figure 1 Production scheduling flowchart for automotive gearbox systems.**

insight: Let $U$ be the set of unscheduled products and $P_U$ be the total processing time of all products in $U$. If a product $T_j$ has no successors in $U$ and has the smallest $f_j(P_U)$-value—i.e., $f_j(P_U) = \min_{T_{j'} \in U} f_{j'}(P_U)$—then $T_j$ should be scheduled at the last position among all products in $U$ to achieve an optimal schedule.

In the following algorithm (Algorithm 1: Algorithm $C_{max}$-$F_{max}$), we adapt the idea of Lawler's algorithm to schedule the unique subassemblies of products in $\mathscr{T}'$ (where $\mathscr{T}'$ consists of $n'$ unscheduled independent products, meaning there are no precedence constraints among them) within a designated time interval $[\tilde{t}, \tilde{t} + \Sigma_{T_{j'} \in \mathscr{T}'} t_{j'}^{(2)})$. For $i = n', n' - 1, \ldots, 1$, we select a product $T_{S(i)} \in U = \mathscr{T}' \backslash \{T_{S(n')}, T_{S(n'-1)}, \ldots, T_{S(i+1)}\}$ such that $f_{S(i)}(\tilde{t} + \Sigma_{T_{j'} \in U} t_{j'}^{(2)}) = \min_{T_j \in U} f_j(\tilde{t} + \Sigma_{T_{j'} \in U} t_{j'}^{(2)})$ and place it as the last product in $U$. The sequence $S(1), S(2), \ldots, S(n')$ represents the indices of the products in $\mathscr{T}'$ which are processed on the facility during the time interval $[\tilde{t}, \tilde{t} + \Sigma_{T_{j'} \in \mathscr{T}'} t_{j'}^{(2)})$. For simplicity, we will refer to this sequence as being scheduled in Lawler's order.

Let us briefly explain the meaning of $\tilde{t}$, which will become clearer in Step 2.2 of Algorithm $C_{max}$-$F_{max}$. The algorithm essentially involves scheduling products in reverse order, moving backward along the time axis (from right to left). Suppose we begin scheduling certain unique subassemblies immediately after the completion of the $i$-th batch of common subassemblies (counting from left to right). In this context, $\tilde{t}$ represents the exact beginning time of the first unique subassembly among those being scheduled.

We will utilize the following well-established method for multicriteria scheduling (*Hoogeveen, 2005*) to identify Pareto optimal schedules.

**Lemma 1** (Hoogeveen, 2005). *Let y be the optimal value for the problem of minimizing $\tau$ under the constraint $\rho \leq \hat{x}$(where $\hat{x}$ is a known upper bound on $\rho$), and let x be the optimal value for the problem of minimizing $\rho$ under the constraint $\tau \leq y$. Then, $(x, y)$ is identified as a Pareto optimal point for two criteria $\rho$ and $\tau$.*

The following lemma outlines the structure of the Pareto optimal schedules that we aim to identify. The proof is omitted here as it closely parallels the one presented in *Gerodimos, Glass & Potts (2000)*.

**Lemma 2**. *For every Pareto optimal point in $1| \prec, 2 - subpt, BA|(C_{max}, f_{max})$, there is an associated schedule where the common subassembly of each product is placed in the batch that directly precedes its unique subassembly.*

According to Lemma 2, a feasible schedule can be expressed as a sequence of *product-subsequences* $Q_1, Q_2, \ldots, Q_n$, where $Q_i$ comprises the products whose common subassemblies are processed in the $i$-th batch. Particularly, we assume that only the last $l$ product-subsequences are nonempty. Hence, it is evident that the subsequences $Q_{n-l+1}, Q_{n-l+2}, \ldots, Q_n$ form a partition of $\mathscr{T}$ and must adhere to the precedence constraints.

The sub-schedule for the products in $Q_i$ can be represented as $\lambda_i \mu_i$, where $\lambda_i$ is a *batch* containing the common subassemblies of the products in $Q_i$, and $\mu_i$ is a *unique-subsequence* comprising the unique subassemblies of the products in $Q_i$ arranged in Lawler's order. Let $t(\lambda_i) = \Sigma_{T_j \in Q_i} t_j^{(1)}$ and $t(\mu_i) = \Sigma_{T_j \in Q_i} t_j^{(2)}$ represent the processing times for the batch and the unique-subsequence of $Q_i$, respectively. The processing time of the entire product-subsequence $Q_i$ is denoted by $t(Q_i) = t(\lambda_i) + t(\mu_i)$. Additionally, the processing time for any empty product-subsequence is considered to be zero. The setup time for batch $\lambda_i$ is denoted by $\delta(\lambda_i)$, which is $\delta$ if $\lambda_i$ is nonempty, and 0 otherwise. The beginning time and completion time for $Q_i$ are denoted by $B(Q_i)$ and $C(Q_i)$, respectively. We have the relationship: $C(Q_i) = B(Q_i) + t(Q_i)$. Therefore, we obtain:

**Lemma 3**. *In a feasible schedule $S = (Q_1, Q_2, \ldots, Q_n)$, the relationship is given by:* $B(Q_1) = \delta(\lambda_1)$, $B(Q_i) = C(Q_{i-1}) + \delta(\lambda_i)$, $i = 2, \ldots, n$.

A natural feasible schedule $S^{(0)} = \{Q_1^{(0)}, Q_2^{(0)}, \ldots, Q_n^{(0)}\}$ can be constructed, where $Q_i^{(0)}$ includes all products (vertices) with an out-degree of zero in the graph $G \backslash \cup_{h=i+1}^n Q_h^{(0)}$, $i = n, n-1, \ldots, 1$. The construction of $S^{(0)}$ from $G$ requires $O(n^2)$ time.

Let $\Gamma(\mathscr{T})$ represent the set of all feasible schedules for $\mathscr{T}$. Our focus is on the subset of schedules in $\Gamma(\mathscr{T})$ that possess the characteristics outlined in Lemmas 2 and 3. Define $\Gamma(\mathscr{T}, y)$ as the set of schedules in $\Gamma(\mathscr{T})$ with a $f_{max}$-value less than $y$. Clearly, $\Gamma(\mathscr{T}, +\infty) = \Gamma(\mathscr{T})$. Let $PO(\mathscr{T})$ represent the Pareto set, which includes all Pareto optimal points, with each point associated with a schedule.

We are ready to present the algorithm for solving $1| \prec, 2 - subpt, BA|(C_{max}, f_{max})$, Algorithm $C_{max}$-$F_{max}$ (Algorithm 1).

---

> **Algorithm 1 (Algorithm Cmax-Fmax).**
>
> **Step 1.** Set $PO(\mathcal{T}) = \varnothing$, $e = 0$, and $y^{(e)} = +\infty$. Let the initial schedule $S^{(0)} = (Q_1^{(0)}, Q_2^{(0)}, \ldots, Q_n^{(0)})$ be the natural feasible schedule described earlier.
>
> **Step 2.** During the $(e+1)$-th round:
>
> Set $y^{(e+1)} = f_{\max}(S^{(e)})$. Adjust $S^{(e)} = (Q_1^{(e)}, Q_2^{(e)}, \ldots, Q_n^{(e)})$ to construct the new schedule
> $S^{(e+1)} = (Q_1^{(e+1)}, Q_2^{(e+1)}, \ldots, Q_n^{(e+1)})$ as follows:
>
> **Step 2.1.** For $i = n, n-1, \ldots, 1$, verify the precedence constraints and the inequality $f_j^{(e)} < y^{(e+1)}$ for each product $T_j$ in $Q_i^{(e)}$. The products in $Q_i^{(e)}$ should be checked in reverse order.
>
> **Case (1).** If a product $T_k \in Q_i^{(e)}$ is found such that at least one of its successors has already been moved into $Q_i^{(e)}$, and if $i = 1$, then set $S^{(e+1)} = \varnothing$ and proceed to Step 3. Otherwise ($i \neq 1$), remove $T_k$ from $Q_i^{(e)}$ and insert it into $Q_{i-1}^{(e)}$, appending $T_k$ to the end of $Q_{i-1}^{(e)}$. Job $T_k$ will not be rechecked when next we are checking $Q_{i-1}^{(e)}$.
>
> **Case (2).** If a product $T_k \in Q_i^{(e)}$ is found where the inequality $f_k^{(e)} < y^{(e+1)}$ is violated, and if $i = 1$ or $T_k$ is the last product in $Q_i^{(e)}$, then set $S^{(e+1)} = \varnothing$ and proceed to Step 3. Otherwise, remove $T_k$ along with all the products in $Q_i^{(e)}$ that are in $Q_i^{(e)}$ but processed earlier than $T_k$, and insert them into $Q_{i-1}^{(e)}$ (following the optimality of Lawler's order). Append these moved products to the end of $Q_{i-1}^{(e)}$. These products will not be rechecked when next we are checking $Q_{i-1}^{(e)}$.
>
> **Step 2.2.** Update the modified schedule $S^{(e)}$ as follows: For $i = 1, 2, \ldots, n$, first process the common subassemblies of the products in $Q_i^{(e)}$ as a batch. Then, process the unique subassemblies of the products in $Q_i^{(e)}$ individually, following Lawler's order. Update the cost for each product in the schedule according to Lemma 3.
>
> **Step 2.3.** Repeat Steps 2.1 and 2.2 until all inequalities and precedence constraints in the modified schedule $S^{(e)}$ are satisfied. Once no violations remain, let $S^{(e+1)}$ be the final modified schedule.
>
> **Step 3.** If $S^{(e+1)} = \varnothing$, then set $PO(\mathcal{T}) = PO(\mathcal{T}) \cup \{(C_{\max}(S^{(e)}), f_{\max}(S^{(e)}), S^{(e)})\}$ and return $PO(\mathcal{T})$. Otherwise, if $C_{\max}(S^{(e+1)}) gt; C_{\max}(S^{(e)})$, then update $PO(\mathcal{T}) = PO(\mathcal{T}) \cup \{(C_{\max}(S^{(e)}), f_{\max}(S^{(e)}), S^{(e)})\}$.
>
> **Step 4.** Set $e = e + 1$ and go to Step 2.

Step 1 of Algorithm $C_{\max}$-$F_{\max}$ can be executed in $O(n^2)$ time. Step 2 requires $O(n^2)$ time for each pass. During each pass of Step 2.1, every product is checked exactly once, starting from the last product-subsequence and proceeding to the first. Multiple passes of Steps 2.1 and 2.2 may occur in a single round. Steps 3 and 4 can be completed in $O(1)$ time per round.

In each pass, at least one product must be moved to the left. It is important to note that in Algorithm 1, products can only be moved leftward. Given that there are $n$ product-subsequences, each product can be moved to the left at most $n - 1$ times. Consequently, the total number of passes is $O(n^2)$. Therefore, the overall running time of Algorithm $C_{\max}$-$F_{\max}$ is $O(n^4)$.

Heuristic methods (*e.g.*, *Hidri & Tlija, 2024*) are typically employed to handle complex dynamic constraints such as sequence-dependent setup times, but they come at the cost of optimality. The exact algorithm proposed above achieves a balance between computational efficiency and solution quality in polynomial time through rational design of batch structures (*e.g.*, batch processing of common subassemblies and scheduling of unique subassemblies in Lawler's order). Compared with heuristic methods, this algorithm provides theoretical optimality guarantees; compared with high-complexity exact methods (such as $O(n^6)$ algorithms), its computational efficiency is significantly improved. Future

research can further expand its applicability in industrial scenarios by integrating dynamic constraint management (*e.g.*, real-time priority adjustment) and parallel computing technologies (*e.g.*, GPU acceleration).

**Lemma 4** *Let $S^{(e)} = (Q_1^{(e)}, Q_2^{(e)}, \ldots, Q_n^{(e)})$ be obtained at the e-th round of Algorithm $C_{max}$-$F_{max}$, where $e \geq 0$. Let $S = (Q_1, Q_2, \ldots, Q_n)$ be any one in $\Gamma(\mathcal{T}, y^{(e)})$. Then, $\cup_{q=i}^n Q_q \subseteq \cup_{q=i}^n Q_q^{(e)}$, $i = n, n-1, \ldots, 1$.*

**Proof 1** *The proof is conducted using induction on e.*

*The base case $e = 0$ is verified trivially. The initial schedule $S^{(0)} = (Q_1^{(0)}, Q_2^{(0)}, \ldots, Q_n^{(0)})$ is the natural feasible schedule, which inherently satisfies the lemma. This is because each product in $Q_i^{(e)}$ must have a successor in $Q_{i+1}^{(e)}$, and therefore, it cannot be included in $Q_{i+1}^{(e)}$ in any schedule in $\Gamma(\mathcal{T}, y^{(0)}) = \Gamma(\mathcal{T})$, $i = 1, 2, \ldots, n-1$.*

*Assume that the lemma holds for $S^{(e)}$ and any schedule in $\Gamma(\mathcal{T}, y^{(e)})$. Now, consider $S^{(e+1)}$ and any schedule $S \in \Gamma(\mathcal{T}, y^{(e+1)})$. Since $y^{(e+1)} < y^{(e)}$, $S \in \Gamma(\mathcal{T}, y^{(e)})$. By the inductive assumption, the lemma holds for $S^{(e)}$ and S. Therefore, for $i = n, n-1, \ldots, 1$, we have $\cup_{q=i}^n Q_q \subseteq \cup_{q=i}^n Q_q^{(e)}$. Equivalently, we have: for $i = 1, 2, \ldots, n$, $\cup_{q=1}^i Q_q^{(e)} \subseteq \cup_{q=1}^i Q_q$.*

*Consider the first (i.e., rightmost) inequality violation during the $(e+1)$-th round. Let the first moved product be $T_k \in Q_i^{(e)}$. Since the inequality $f_k^{(e)} < y^{(e+1)}$ does not hold and the unique subassemblies of the products in $Q_i^{(e)}$ are scheduled in Lawler's order, $T_k$ and all the products which are in $Q_i^{(e)}$ but scheduled earlier than $T_k$ cannot stay in $\cup_{q=i}^n Q_q^{(e)}$. Therefore, we move these products from $Q_i^{(e)}$ into $Q_{i-1}^{(e)}$. By the inductive assumption, in S any of these products cannot stay in $\cup_{q=i}^n Q_q$, otherwise the last one will complete no earlier than $C_k(S^{(e)})$ and thus incur an inequality violation.*

*After moving these products to the left in $S^{(e)}$, we further adjust $S^{(e)}$ to obey the precedence constraints. Observing S accordingly, we can know that S coincides with the adjustment. Then we consider the next inequality violation, and so on. By repeating this argument, we ultimately demonstrate that the lemma holds for the $(e+1)$-th round.*

*By applying the principle of induction, the proof is thereby completed.*
    We get:

**Lemma 5** *Let $S^{(e)} = (Q_1^{(e)}, Q_2^{(e)}, \ldots, Q_n^{(e)})$ be obtained at the e-th round of Algorithm $C_{max}$-$F_{max}$, where $e \geq 0$. Let $S = (Q_1, Q_2, \ldots, Q_n)$ be any one in $\Gamma(\mathcal{T}, y^{(e)})$. Then:*
    *(1) $l(S^{(e)}) \leq l(S)$, where $l(S^{(e)})$ and $l(S)$ represent the number of nonempty product-subsequences in $S^{(e)}$ and S respectively;*
    *(2) $B(Q_i^{(e)}) \leq B(Q_i)$, $i = 1, 2, \ldots, n$;*
    *(3) $C(Q_i^{(e)}) \leq C(Q_i)$, $i = 1, 2, \ldots, n$.*

**Lemma 6** *Let $S^{(e)}$ be obtained at the e-th round of Algorithm $C_{max}$-$F_{max}$, where $e \geq 0$. If $S^{(e)} = \varnothing$, then $\Gamma(\mathcal{T}, y^{(e)}) = \varnothing$. Otherwise, $S^{(e)}$ has minimum makespan in all schedules in $\Gamma(\mathcal{T}, y^{(e)})$.*

**Proof 2** *In the implementation of Algorithm $C_{max}$-$F_{max}$, we will get $S^{(e)} = \varnothing$ when one of the following two cases occurs:* **Case (1).** *Find a product $T_k \in Q_i^{(e-1)}$ such that at least one of its successors has already been moved into $Q_i^{(e-1)}$ and $i = 1$.* **Case (2).** *Find a product $T_k \in Q_i^{(e-1)}$ where $f_k^{(e-1)} < y^{(e)}$ is violated, and either $i = 1$ or $T_k$ is the last product in $Q_i^{(e-1)}$. In Case (1), we know that $T_k$ cannot be scheduled without violating the precedence constraints in any feasible schedule. Therefore, we set $S^{(e)} = \varnothing$ indicating that $\Gamma(\mathcal{T}, y^{(e)}) = \varnothing$. In Case (2), if $i = 1$, we know that no feasible schedule can arrange $T_k$ with a cost less than $y^{(e)}$. If $T_k$ is the last product in $Q_i^{(e-1)}$, then there will be an empty product-subsequence between two nonempty product-subsequences. We do not need to consider such a case. Therefore, we also set $S^{(e)} = \varnothing$ indicating that $\Gamma(\mathcal{T}, y^{(e)}) = \varnothing$.*

*Conversly, if $S^{(e)} \neq \varnothing$, then by Lemma 5, $S^{(e)}$ has minimum makespan in all schedules in $\Gamma(\mathcal{T}, y^{(e)})$.*

By integrating the results of Lemmas 1 and 6, we obtain the following result:

**Theorem 1** *Algorithm $C_{max}$-$F_{max}$ solves problem $1| \prec, 2 - subpt, BA|(C_{max}, f_{max})$ in $O(n^4)$ time. Moreover, the final schedule generated by the algorithm has the minimum makespan in all optimal schedules for $1| \prec, 2 - subpt, BA|f_{max}$.*

To solve $1| \preceq, 2 - subpt, BA|(C_{\max}, f_{\max})$ (the weak precedence constraints), we need to modify Step 1, Step 2.1 and Step 2.2 of Algorithm $C_{\max}$-$F_{\max}$ slightly. In Step 1, we set the initial schedule $S^{(0)} = (\varnothing, \varnothing, \ldots, \varnothing, \mathcal{T})$, where the common subassemblies of the products in $\mathcal{T}$ are first processed as a batch, followed by the unique subassemblies, which are processed individually according to Lawler's order while adhering to the precedence constraints. That is, we use the actual Lawler's algorithm (*Lawler, 1973*) to schedule the unique subassemblies of the products in $\mathcal{T}$ in the time interval $[\delta + \Sigma_{T_{j'} \in \mathcal{T}} t_{j'}^{(1)}, \delta + \Sigma_{T_{j'} \in \mathcal{T}} t_{j'}^{(1)} + \Sigma_{T_{j'} \in \mathcal{T}} t_{j'}^{(2)})$. In Step 2.1, since the common subassemblies of $T_k$ and its successors can be in the same batch, we do not need Case (1). We just need Case (2) to ensure that the predecessors of $T_k$ and $T_k$ obey the weak precedence constraints. In Step 2.2, the unique subassemblies of the products in $Q_i^{(e)}$ must be processed individually, following Lawler's order and complying with the precedence constraints.

Then we get:

**Theorem 2** *Modified Algorithm $C_{max}$-$F_{max}$ solves problem $1| \preceq, 2 - subpt, BA|(C_{max}, f_{max})$ in $O(n^4)$ time. Moreover, the final schedule generated by the algorithm has the minimum makespan in all optimal schedules for $1| \preceq, 2 - subpt, BA|f_{max}$.*

## THE EXPERIMENTS

In this section, we present experimental results of our Algorithm $C_{\max}$-$F_{\max}$. Next, we will compare with algorithms for Pareto scheduling two-subassembly products without precedence constraints. The algorithm, implemented in PyCharm, was tested on randomly generated instances. We varied key factors. The number of jobs $n \in (10, 100)$. The processing time for the common subassembly is uniformly distributed over the integer range $[1, 5]$, and the processing time for the unique subassembly is uniformly distributed

**Table 1 Computational times of Algorithm $C_{max}$-$F_{max}$ with precedence constraints.**

| Number of jobs | Average-time (s) | Max-times (s) |
|---|---|---|
| 10 | 2.107859e−04 | 2.729893e−04 |
| 20 | 7.882357e−04 | 8.480549e−04 |
| 30 | 1.990342e−03 | 2.228498e−03 |
| 40 | 4.230165e−03 | 7.481098e−03 |
| 50 | 6.762409e−03 | 7.446527e−03 |
| 60 | 1.205318e−02 | 1.517057e−02 |
| 70 | 1.542490e−02 | 1.956534e−02 |
| 80 | 2.106686e−02 | 2.891660e−02 |
| 90 | 3.026404e−02 | 3.747702e−02 |
| 100 | 4.073451e−02 | 5.063152e−02 |

**Table 2 Computational times of Algorithm $C_{max}$-$F_{max}$ without precedence constraints.**

| Number of jobs | Average time (s) | Max time (s) |
|---|---|---|
| 10 | 1.636958E−03 | 2.530813E−03 |
| 20 | 1.394284E−02 | 1.494074E−02 |
| 30 | 5.668330E−02 | 5.867529E−02 |
| 40 | 1.624435E−01 | 1.744251E−01 |
| 50 | 3.673631E−01 | 3.688035E−01 |
| 60 | 7.248310E−01 | 7.265387E−01 |
| 70 | 1.296956E+00 | 1.299667E+00 |
| 80 | 2.157462E+00 | 2.162263E+00 |
| 90 | 3.386713E+00 | 3.426600E+00 |
| 100 | 5.072271E+00 | 5.111201E+00 |

over the integer range [1, 10]. To implement the algorithm, 10 experiments were conducted for each order of magnitude of the job.

The average and maximum running times of Algorithm 1 in the article are shown in Table 1. In Table 2, we present the running times of the problem $1|2 - subpt, BA|(C_{max}, f_{max})$. To more effectively demonstrate the variation in average execution time relative to the number of jobs across both experimental conditions, the temporal performance trends are visualized in Figs. 2 and 3. using linear graphical representations.

Figure 2 demonstrates Algorithm $C_{max}$-$F_{max}$ performance With precedence constraints. This figure compares the average and maximum execution times of Algorithm $C_{max}$-$F_{max}$ under precedence constraints. The x-axis represents the number of jobs (ranging from 10 to 100), while the y-axis shows the runtime in seconds. The linear trend demonstrates that the algorithm's runtime scales polynomially with problem size, as expected from its theoretical $O(n^4)$ complexity.

Figure 3 demonstrates Algorithm $C_{max}$-$F_{max}$ performance without precedence constraints. Similar to Fig. 1, this figure plots the runtime of Algorithm 1 but excludes

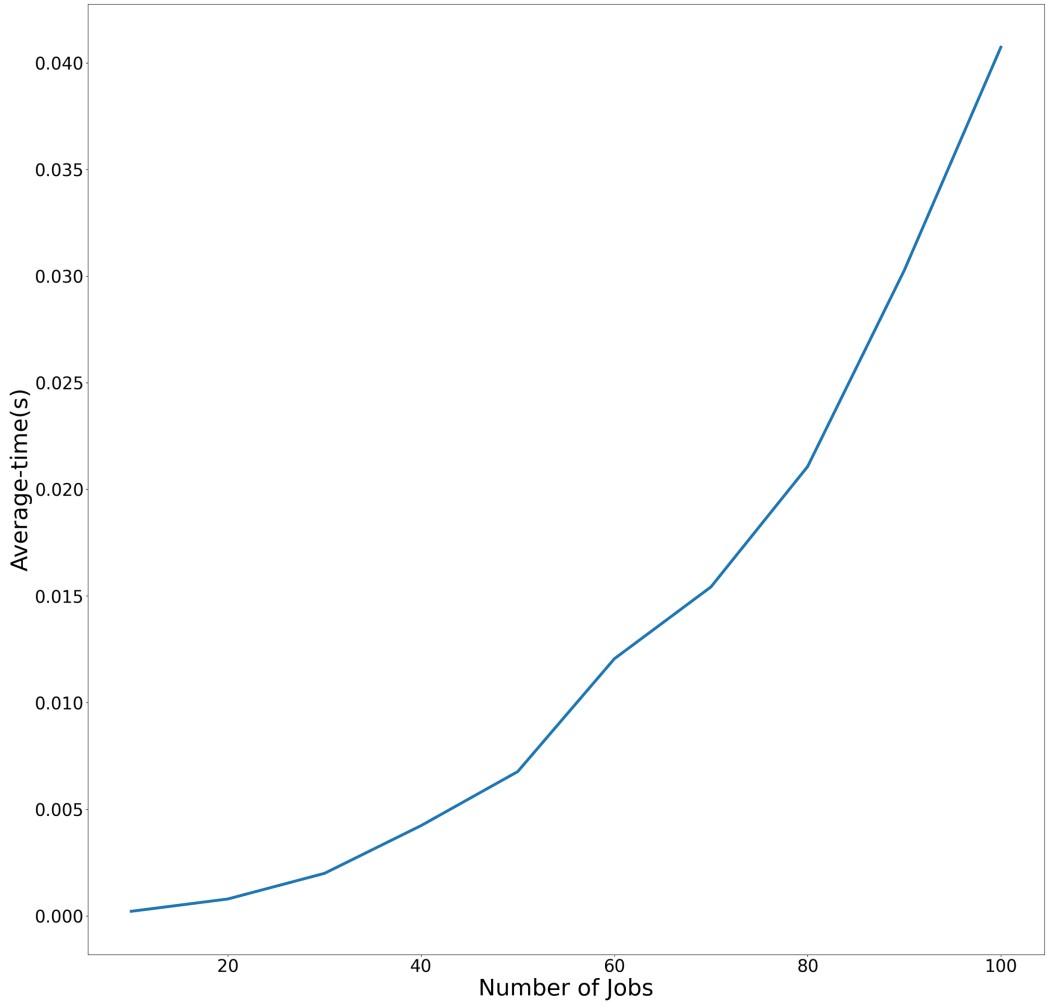

**Figure 2 Average-time trend of Algorithm $C_{max}$-$F_{max}$ with precedence constraints as the number of jobs increases.**

precedence constraints. The steeper slope of the curves indicates that Algorithm $C_{max}$-$F_{max}$ with precedence constraints significantly reduce computational overhead.

The computational efficiency comparison between Algorithm $C_{max}$-$F_{max}$ (with precedence constraints) and Algorithm $C_{max}$-$F_{max}$ (without precedence constraints) is quantified in Fig. 4. This area chart tracks the growing performance gap as job size ($n$) increases from 10 to 100. Key observations include: At $n = 10$, the time difference is minimal (0.0014 s). The gap grows polynomially, reaching 5.03 s at $n = 100$. This visualization demonstrates how Algorithm 1's strategic batch scheduling under precedence constraints consistently outperforms traditional approaches, particularly for $n > 50$ where the difference becomes operationally significant in real-world scheduling scenarios.

The significance of algorithm running time extends beyond mere efficiency; it also serves as a quantitative indicator of theoretical solvability, thereby determining whether the model can transition from a theoretical framework to practical implementation.

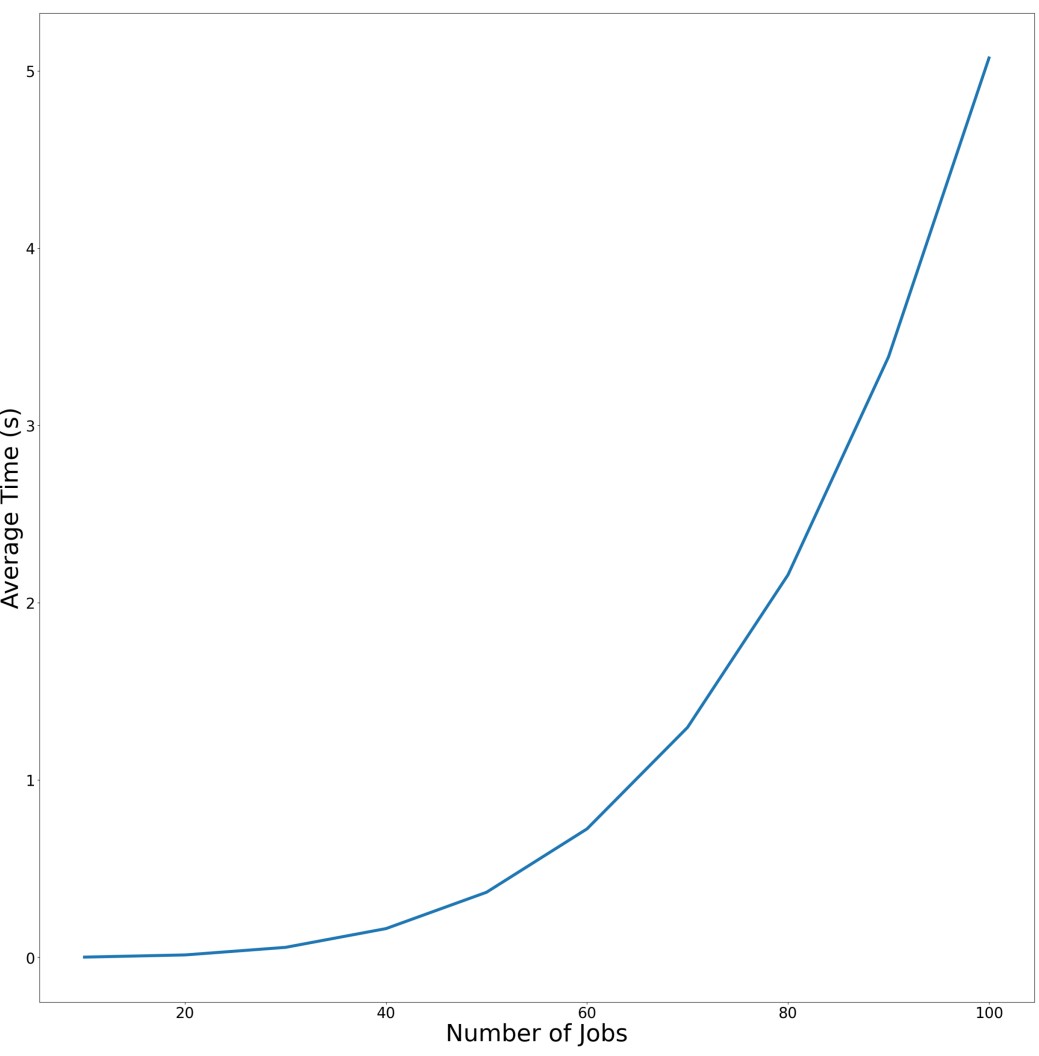

**Figure 3  Average-time trend of Algorithm $C_{max}$-$F_{max}$ without precedence constraints as the number of jobs increases.**               

For real-world instances ($n \leq 100$), $O(n^4)$ ensures solvability within seconds. Compared to others, slower algorithms (*e.g.*, $O(n^6)$ for bounded batch scheduling *He, Lin & Lin (2015)*) become infeasible.

## ALGORITHMS FOR LEXICOGRAPHICAL SCHEDULING TWO-SUBASSEMBLY PRODUCTS WITH PRECEDENCE CONSTRAINTS

Existing exact methods mostly focus on bi-objective optimization, while this article first achieves three-objective lexicographical optimization ($f_{max}, g_{max}, C_{max}$), which is closer to the multi-dimensional optimization needs in practical manufacturing (such as considering delay penalties, resource costs, and makespan simultaneously).

In this section, we focus on presenting an $O(n^4)$-time algorithm designed to solve problem $1| \prec, 2 - subpt, BA|Lex(f_{max}, g_{max}, C_{max})$. Towards the end of this section, we will

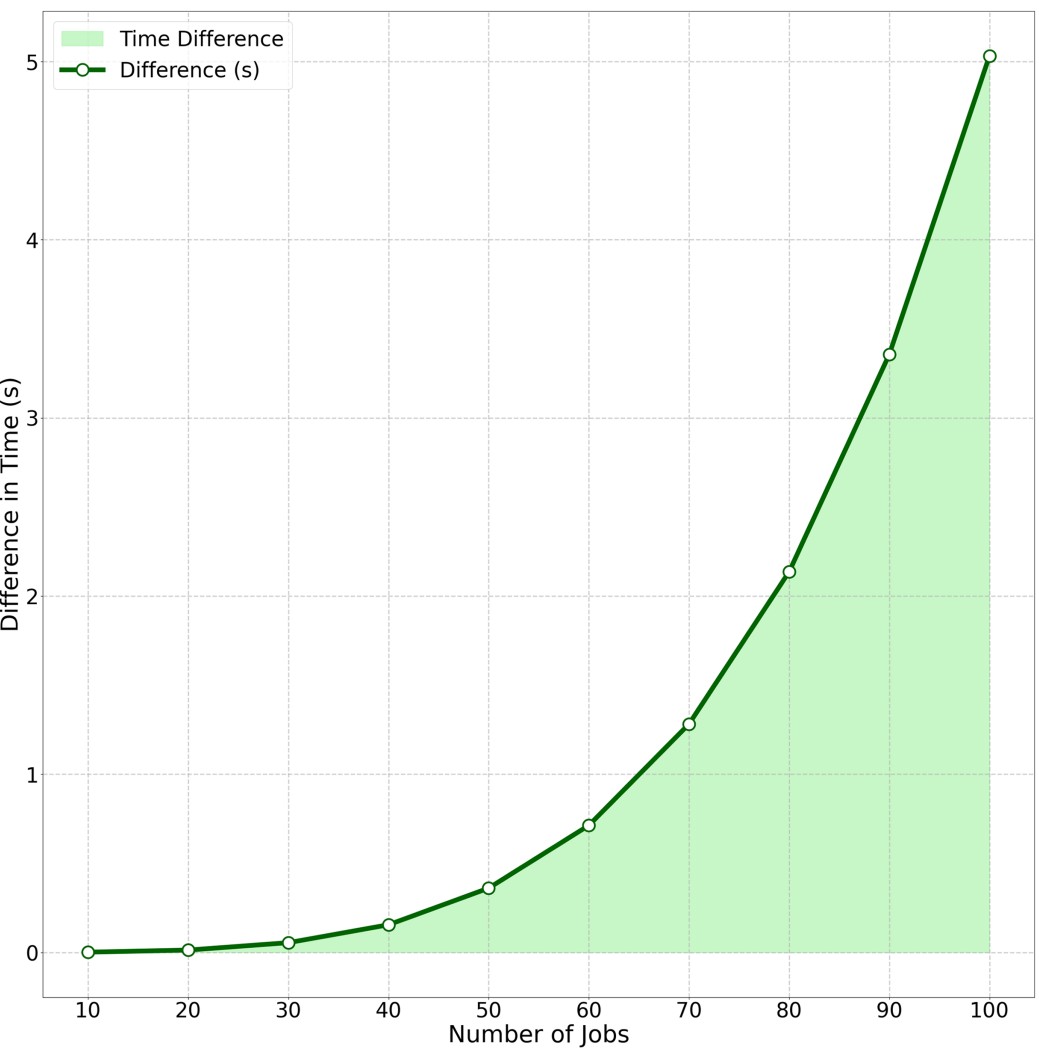

**Figure 4** **Average time difference between algorithm $C_{max}$-$F_{max}$ (With precedence constraints) and algorithm $C_{max}$-$F_{max}$ (Without precedence constraints).**

demonstrate how slight modifications to the algorithm enable it to also solve problem $1| \preceq, 2 - subpt, BA|Lex(f_{max}, g_{max}, C_{max})$ in $O(n^4)$ time.

Let $S^*$ represent the final schedule produced by Algorithm $C_{max}$-$F_{max}$. According to Theorem 1, $S^*$ achieves the minimum makespan in all optimal schedules for problem $1| \prec, 2 - subpt, BA|f_{max}$.

Let $\Gamma(\mathscr{T}, f_{max}(S^*))$ denote the set of feasible schedules for $\mathscr{T}$ whose $f_{max}$-values are equal to $f_{max}(S^*)$. It is important to note that Lemma 2 (with "Pareto optimal" replaced by "lexicographically optimal") and Lemma 3 still apply to $1| \prec, 2 - subpt, BA|Lex(f_{max}, g_{max}, C_{max})$. Therefore, our focus is on the schedules in $\Gamma(\mathscr{T}, f_{max}(S^*))$ that exhibit the properties described in Lemmas 2 and 3. Let $\Gamma(\mathscr{T}, f_{max}(S^*), Y)$ represent the set of schedules in $\Gamma(\mathscr{T}, f_{max}(S^*))$ whose $g_{max}$-values are less than $Y$. Consequently, we have $\Gamma(\mathscr{T}, f_{max}(S^*), +\infty) = \Gamma(\mathscr{T}, f_{max}(S^*))$.

---

**Algorithm 2 (Algorithm Fmax-Gmax-Cmax).**

***Step 1.*** *Set $e = 0$ and $Y^{(e)} = +\infty$. Let the initial schedule $S^{(0)} = (Q_1^{(0)}, Q_2^{(0)}, \ldots, Q_n^{(0)}) = S^*$, where $S^*$ is the last schedule generated by Algorithm $C_{max}$-$F_{max}$. The products in $Q_i^{(0)}$ are stored in a cyclic queue, $i = 1, 2, \ldots, n$.*

***Step 2.*** *During the $(e + 1)$-th round:*

*Set $Y^{(e+1)} = g_{\max}(S^{(e)})$. Adjust $S^{(e)} = (Q_1^{(e)}, Q_2^{(e)}, \ldots, Q_n^{(e)})$ to construct the new schedule $S^{(e+1)} = (Q_1^{(e+1)}, Q_2^{(e+1)}, \ldots, Q_n^{(e+1)})$ as follows:*

***Step 2.1.*** *For $i = n, n-1, \ldots, 1$, for each product $T_j$ in $Q_i^{(e)}$, check both the precedence constraints and the two inequalities $f_j^{(e)} \leq f_{\max}(S^*)$ and $g_j^{(e)} < Y^{(e+1)}$. The products in $Q_i^{(e)}$ are checked in reverse order, starting from the end of the queue.*

***Case (1).*** *If a product $T_k \in Q_i^{(e)}$ is found where at least one of its successors has already been moved into $Q_i^{(e)}$, and if $i = 1$, then set $S^{(e+1)} = \varnothing$ and proceed to Step 3. Otherwise ($i \neq 1$), remove $T_k$ from $Q_i^{(e)}$ and insert it into $Q_{i-1}^{(e)}$, appending it to the end of $Q_{i-1}^{(e)}$. Job $T_k$ will not be rechecked when next we are checking $Q_{i-1}^{(e)}$.*

***Case (2).*** *If a product $T_k \in Q_i^{(e)}$ is found where either $f_k^{(e)} \leq f_{\max}(S^*)$ or $g_k^{(e)} < Y^{(e+1)}$ is violated, let $E_k(S^{(e)})$ denote the set of products that are in $Q_i^{(e)}$ but processed earlier than $T_k$ in $S^{(e)}$. Let $SE_k(S^{(e)})$ represent the set of suitable earlier products of $T_k$. That is, $SE_k(S^{(e)}) = \{T_j \in E_k(S^{(e)}) | f_j(C_k(S^{(e)})) \leq f_{\max}(S^*) \wedge g_j(C_k(S^{(e)})) < Y^{(e+1)}\}$. We then distinguish between two different subcases:*

***Subcase (2.1).*** *$SE_k(S^{(e)}) \neq \varnothing$.*

*Pick a product in $SE_k(S^{(e)})$ and let it be scheduled immediately after $T_k$ in the same product-subsequence. Thus, this product is completed exactly at $C_k(S^{(e)})$.*

***Subcase (2.2).*** *$SE_k(S^{(e)}) = \varnothing$.*

*If $i = 1$ or $T_k$ is the last scheduled product in $Q_i^{(e)}$, then set $S^{(e+1)} = \varnothing$ and proceed to Step 3. Otherwise, (DEQUEUE) remove $T_k$ along with all the products in $E_k(S^{(e)})$, and (ENQUEUE) insert these products into $Q_{i-1}^{(e)}$. These products will not be rechecked when next we are checking $Q_{i-1}^{(e)}$.*

***Step 2.2.*** *Update the modified schedule $S^{(e)} = (Q_1^{(e)}, Q_2^{(e)}, \ldots, Q_n^{(e)})$ as follows: For $i = 1, 2, \ldots, n$, first process the common subassemblies of the products in $Q_i^{(e)}$ as a batch. Then, process the unique subassemblies of the products in $Q_i^{(e)}$ individually in the sorted order (which may differ from Lawler's order). Update the two maximum costs for each product in the schedule according to Lemma 3.*

***Step 2.3.*** *Repeat Steps 2.1 and 2.2 until all inequalities and precedence constraints in the modified schedule $S^{(e)}$ are satisfied. Once no violations remain, let $S^{(e+1)}$ be the final modified schedule.*

***Step 3.*** *If $S^{(e+1)} \neq \varnothing$, then set $e = e + 1$ and go to Step 2. Otherwise, return $S^{(e)}$.*

---

We are ready to present the algorithm for solving $1 | \prec, 2 - subpt, BA | Lex(f_{\max}, g_{\max}, C_{\max})$, Algorithm $F_{\max}$-$G_{\max}$-$C_{\max}$ (Algorithm 2).

For further details on the cyclic queue data structure and its basic operations, DEQUEUE and ENQUEUE, the reader may refer to *Cormen et al. (2022)*.

Similar to the time complexity analysis of Algorithm $C_{\max}$-$F_{\max}$, it can be demonstrated that the running time of Algorithm $F_{\max}$-$G_{\max}$-$C_{\max}$ is $O(n^4)$.

We get:

**Lemma 7** *Let $S^{(e)} = (Q_1^{(e)}, Q_2^{(e)}, \ldots, Q_n^{(e)})$ be obtained at the e-th round of Algorithm $F_{max}$-$G_{max}$-$C_{max}$, where $e \geq 0$. Let $S = (Q_1, Q_2, \ldots, Q_n)$ be any one in $\Gamma(\mathcal{T}, f_{max}(S^*), Y^{(e)})$. Then, $\cup_{q=i}^{n} Q_q \subseteq \cup_{q=i}^{n} Q_q^{(e)}$, $i = n, n-1, \ldots, 1$.*

**Lemma 8** *Let $S^{(e)} = (Q_1^{(e)}, Q_2^{(e)}, \ldots, Q_n^{(e)})$ be obtained at the e-th round of Algorithm $F_{max}$-$G_{max}$-$C_{max}$, where $e \geq 0$. Let $S = (Q_1, Q_2, \ldots, Q_n)$ be any one in $\Gamma(\mathcal{T}, f_{max}(S^*), Y^{(e)})$. Then:*

**Table 3 Computational times of algorithm F$_{\text{max}}$-G$_{\text{max}}$-C$_{\text{max}}$.**

| Number of jobs | Average time (s) | Max time (s) |
|---|---|---|
| 10 | 5.454063E−04 | 2.706051E−03 |
| 20 | 1.091480E−03 | 2.664328E−03 |
| 30 | 2.777719E−03 | 9.774208E−03 |
| 40 | 1.028388E−02 | 5.631423E−02 |
| 50 | 9.011396E−03 | 5.189085E−02 |
| 60 | 1.748489E−02 | 7.746744E−02 |
| 70 | 1.449437E−02 | 4.745579E−02 |
| 80 | 1.357992E−02 | 2.614021E−02 |
| 90 | 3.757528E−02 | 1.367950E−01 |
| 100 | 2.615252E−02 | 8.715153E−02 |

**Table 4 Detailed difference.**

| Aspect | Algorithm C$_{\text{max}}$-F$_{\text{max}}$ | Algorithm F$_{\text{max}}$-G$_{\text{max}}$-C$_{\text{max}}$ |
|---|---|---|
| Objective | Bicriteria Pareto optimization: $\min(C_{max}, f_{max})$ | Lexicographic tri-criteria optimization: $\min f_{max} \rightarrow \min g_{max} \rightarrow \min C_{max}$ |
| Theoretical basis | Lemma 2 (Batch adjacency property) | Theorem 4 (Lexicographic optimality) |
| Output | Pareto frontier $PO(\mathscr{T})$ | Optimal schedule $S_{\text{lex}}$ |

(1) $l(S^{(e)}) \leq l(S)$, where $l(S^{(e)})$ and $l(S)$ represent the number of nonempty product-subsequences in $S^{(e)}$ and $S$ respectively;

(2) $B(Q_i^{(e)}) \leq B(Q_i)$, $i = 1, 2, \ldots, n$;

(3) $C(Q_i^{(e)}) \leq C(Q_i)$, $i = 1, 2, \ldots, n$.

**Lemma 9** *Let $S^{(e)}$ be obtained at the e-th round of Algorithm $F_{max}$-$G_{max}$-$C_{max}$, where $e \geq 0$. If $S^{(e)} = \varnothing$, then $\Gamma\left(\mathscr{T}, f_{max}(S^*), Y^{(e)}\right) = \varnothing$. Otherwise, $S^{(e)}$ has minimum makespan in all schedules in $\Gamma\left(\mathscr{T}, f_{max}(S^*), Y^{(e)}\right)$.*

**Theorem 3** *Algorithm $F_{max}$-$G_{max}$-$C_{max}$ solves $1| \prec, 2-\text{subpt}, BA|Lex(f_{max}, g_{max}, C_{max})$ in $O(n^4)$ time.*

To solve $1| \preceq, 2 - subpt, BA|Lex(f_{\max}, g_{\max}, C_{\max})$, we need to modify Step 1 and Step 2.1 of Algorithm F$_{\text{max}}$-G$_{\text{max}}$-C$_{\text{max}}$ slightly. In Step 1, we set the initial schedule $S^{(0)} = S^*$, where $S^*$ is the last schedule generated by modified Algorithm C$_{\text{max}}$-F$_{\text{max}}$. In Step 2.1, since the common subassemblies of $T_k$ and its successors can be in the same batch, we do not need Case (1). We just need Case (2) to ensure that the predecessors of $T_k$ and $T_k$ obey the weak precedence constraints.

Then we get:

**Theorem 4** *Modified Algorithm $F_{max}$-$G_{max}$-$C_{max}$ solves $1| \preceq, 2 - subpt, BA|Lex(f_{max}, g_{max}, C_{max})$ in $O(n^4)$ time.*

For Algorithm F$_{\text{max}}$-G$_{\text{max}}$-C$_{\text{max}}$, we perform the same experiment as in "The Experiments". The running time of problem $1| \prec, 2 - subpt, BA|Lex(f_{\max}, g_{\max}, C_{\max})$ is shown in Table 3.

Table 4 systematically compares two scheduling algorithms, Algorithm $C_{max}$-$F_{max}$ and Algorithm $F_{max}$-$G_{max}$-$C_{max}$, thereby highlighting their distinct optimization paradigms, theoretical underpinnings, and outputs.

Algorithm $C_{max}$-$F_{max}$ targets bicriteria Pareto optimization (minimizing $C_{max}$ and $f_{max}$ simultaneously) and leverages Lemma 2 to simplify batch sequencing, yielding a Pareto frontier $PO(\mathcal{T})$ of non-dominated solutions. In contrast, Algorithm $F_{max}$-$G_{max}$-$C_{max}$ enforces lexicographic tri-criteria priority ($f_{max} \rightarrow g_{max} \rightarrow C_{max}$) grounded in Theorem 4, producing a optimal schedule $S_{lex}$. These design choices reflect trade-offs between exploring solution diversity ($C_{max}$-$F_{max}$) and enforcing strict priority ($F_{max}$-$G_{max}$-$C_{max}$), enabling application-specific deployment.

## CONCLUSIONS

In this article, we explored four multicriteria scheduling problems involving two-subassembly products with precedence constraints on a fabrication facility, assuming batch availability of the common subassemblies. We introduced an $O(n^4)$-time algorithm for the simultaneous optimization of makespan and maximum cost under both strict and weak precedence constraints. Additionally, we proposed an $O(n^4)$-time algorithm for the lexicographical optimization of two maximum costs and makespan, also under strict or weak precedence constraints. Future research could focus on developing algorithms with improved time complexity for these scheduling problems. A particularly intriguing direction would be to explore Pareto optimization for a general min-max objective function, in conjunction with a general min-max or min-sum objective function.

This study provides a foundational framework for multicriteria scheduling of two-subassembly products, but several promising avenues for extension exist, particularly those that balance theoretical rigor with industrial applicability:

(1) Parallelization for industrial-scale instances. Given the $O(n^4)$ time complexity of the proposed algorithms, developing GPU-accelerated or distributed (MapReduce) implementations represents a high-impact direction. Such optimizations would bridge the gap to real-world manufacturing scenarios with large product portfolios (*e.g.*, automotive subassembly lines), where real-time scheduling is critical.

(2) Dynamic precedence constraint management. Extending the model to handle real-time updated precedence graphs (*e.g.*, in flexible manufacturing systems) is vital for Industry 4.0 applications. *Xu et al. (2024)* proposed an incremental scheduling algorithm with $O(n^3)$ complexity per update, which could be integrated with our Pareto optimization framework to accommodate dynamic priority changes (*e.g.*, rush orders or machine failures). This enhancement would improve adaptability in volatile production environments.

(3) Energy-aware multicriteria optimization. Incorporating energy consumption as a tertiary objective aligns with sustainability trends in manufacturing. Recent work by *Hidri & Tlija (2024)* shows that dual-objective algorithms can be extended to three criteria with only polynomial complexity growth. For example, minimizing energy use during batch setup ($\delta$) or unique subassembly processing ($t_j^{(2)}$) could be integrated into the

lexicographical optimization framework, balancing makespan, cost, and environmental impact.

These directions are based on the theoretical foundation of this research and also address the unmet demands in industrial scheduling, ensuring the continuous relevance in both academic research and practical applications.

### Funding
This work was supported by Natural Science Foundation of Shandong Province China (No. ZR2020MA030). The funders had no role in study design, data collection and analysis, decision to publish, or preparation of the manuscript.

### Grant Disclosures
The following grant information was disclosed by the authors:
Natural Science Foundation of Shandong Province China: ZR2020MA030.

### Competing Interests
The authors declare that they have no competing interests.

### Author Contributions
- Zhenxin Wen conceived and designed the experiments, performed the experiments, analyzed the data, performed the computation work, prepared figures and/or tables, authored or reviewed drafts of the article, and approved the final draft.
- Shuguang Li conceived and designed the experiments, performed the experiments, analyzed the data, performed the computation work, prepared figures and/or tables, authored or reviewed drafts of the article, and approved the final draft.

### Data Availability
The data is available at GitHub and Zenodo:
- https://github.com/auspicious123123/Multicriteria-scheduling.
- 2023420176@sdtbu.edu.cn. (2025). auspicious123123/Multicriteria-scheduling: Code releases3 (v3.0.0). Zenodo. https://doi.org/10.5281/zenodo.15743494.

### Supplemental Information
Supplemental information for this article can be found online at http://dx.doi.org/10.7717/peerj-cs.3093#supplemental-information.

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
