# Peer review of "Multicriteria scheduling of two-subassembly products with batch availability and precedence constraints"

_PeerJ Computer Science, doi:10.7717/peerj-cs.3093_

## Round 0.1 · original submission · Major Revisions

Thank you for submitting your manuscript "Multicriteria scheduling of two-subassembly products with batch availability and precedence constraints" to PeerJ Computer Science. Based on the reviewers' evaluations, I am recommending that the manuscript undergo major revisions before it can be reconsidered for publication.

While the paper presents rigorous theoretical contributions and proposes polynomial-time algorithms for complex scheduling problems, both reviewers identified significant areas for improvement. These include the need to better contextualize the problem with real-world examples, update and expand the literature review, clarify the interpretation of cost functions and runtime relevance, strengthen the experimental validation, improve the presentation of the algorithms and figures, and address various language and formatting issues.

I encourage you to revise the manuscript thoroughly and provide a detailed response to each reviewer comment upon resubmission. We look forward to evaluating your revised version.

Reviewer 1 ·

Basic reporting

The manuscript addresses an important area in scheduling theory. The authors propose O(n⁴)-time algorithms to solve several scheduling problems considering makespan, maximum cost, and their lexicographical combinations. The paper is methodologically rigorous and presents theoretical developments that are novel and relevant to the scheduling community. However, several important improvements are necessary to enhance the clarity, usability, and impact of the work. My comments are as follows:
- The mathematical modeling - Use a real-world analogy or industrial example to ground the scheduling problem, like manufacturing of modular smartphones, aerospace components, or similar, to aid understanding by a broader audience.
- The literature review section could be expanded to cover more recent studies on scheduling problems and decision-making models. Some references in the literature review are outdated. Including more recent studies would strengthen the manuscript's relevance. For example, Badi et al, 2025. Vendor Managed Inventory in Practice: Efficient Scheduling and Delivery Optimization. Spectrum of Decision Making and Applications.
- The paper defines two regular cost functions for each product (fj and gj). Explain what these cost functions represent in practice. Explain how these costs might correspond to real-world metrics such as penalties for lateness, resource usage, or similar.
- The experiments are synthetic and lack application to a real-world and benchmark dataset. I strongly suggest including a case study or simulated industrial data to demonstrate the practical utility of the proposed algorithms.
- Clarify why the algorithm’s runtime is important given the problem's theoretical nature.
- The algorithms are presented without clearly formatted pseudocode and flow diagrams. Move algorithm descriptions into structured pseudocode blocks and include flowcharts where appropriate.
- Highlight the differences between the original and modified versions of Cmax-Fmax and Fmax-Gmax-Cmax.
- There is no performance comparison with heuristic or exact methods from the literature. Add comparisons to demonstrate the superiority and trade-offs of your approach.
- The manuscript includes many lemmas whose proofs are omitted and overly technical, without adding intuitive value to the reader. My suggestion is to simplify the writing where possible and shift lengthy mathematical derivations to an appendix.
- Emphasize key theoretical insights in the main body.
- Figures in the experimental section, like Figures 1–2, lack proper labels, legends, and explanatory captions.
- The discussion of future work - Be more specific about which directions are most promising and feasible in the research.

Experimental design

See my above report.

Validity of the findings

See my above report.

Reviewer 2 ·

Basic reporting

In general, the English language used throughout the paper is clear and unambiguous. Some suggestions for improving the text:
Pg 9. Line 317. The authors write: “The numbe of jobs n∈(10,100).” Please write “number” instead “numbe”
Pg 9. Lines 317-319. The authors write the following text with some issues.
“The common subassembly processing time uniformly distributed as integers from 1 to 5 and the unique subassembly processing time uniformly distributed as integers from1 to10.”
Missing auxiliary verb — "is" or "are" should be used to complete the passive construction.
"uniformly distributed as integers" is an awkward and incorrect phrasing.
The sentence lacks parallel structure, which makes it harder to read.
It should be clearer that the distributions are uniform over integer ranges.
Pg 9 line 321. The authors write:
"The average and maximum running times of Algorithm Cmax-Fmax in the paper is showed inTable1."
The above sentence has several issues.
Grammar Issues Identified:
(i) Incorrect: "is showed" Fix: "are" instead of "is".
(ii) Fix: Use "shown", which is the correct past participle in passive voice ("are shown").
The corrected version:
"The average and maximum running times of Algorithm Cmax-Fmax in the paper are shown in Table 1."
Pg 12. Lines 383 and 386. Please write “Lemma 7” and “Lemma 8” instead “lemma 7” and “lemma 8” Make this correction in the paper. Write “Proof” when you make a proof. Not “proof”

Cited references are relevant and connected with the content of the paper.

Figures included in the paper are relevant and of high quality. Some suggestions for improving the text:
Pages 11-13. The three figures do not have explanations in the text. They have only a legend with very laconic explanations. I ask the authors to include in the text detailed explanations on each figure. The explanation for Figure 1 is “with precedence constraints”. This is too laconic. The same thing can be said about Figure 2. The explanation of Figure 3 is the following:
“Figure 3. Difference in Average Time Between Two Algorithms. The area chart clearly shows the overall trend of the gap between the two algorithms.”
Better write "Average Time Difference Between Algorithm A and Algorithm B". You should include the names of algorithms A and B.
I suggest that the authors remove the text "The area chart clearly shows" from the legend of Figure 3. Also, I suggest the following text be introduced in the paper body:
“The area chart from Figure 3 illustrates how the time difference between Algorithm A and Algorithm B has evolved over time, highlighting a consistent trend.”

Experimental design

A key contribution of the paper is the development of polynomial-time (O(n4) algorithms that achieve simultaneous optimization of multiple scheduling objectives: specifically, the makespan and the maximum cost. Furthermore, the authors extend their approach to handle lexicographic optimization involving two maximum costs and the makespan, again under both strict and weak precedence conditions. These algorithmic results demonstrate strong theoretical guarantees and provide practical insights for multi-criteria scheduling under complex resource constraints.
The subject of the paper fits with the aim of the journal PeerJ Computer Science.
Research questions are well defined, relevant, and meaningful.
Methods are described with sufficient detail & information. Some suggestions for improving the text:
Pg 9. Line 314. The authors write: “Next, we with compare…”. Probably the authors should replace “with” with “will”.
Also, as it is written, the authors claim that they will compare “experimental results of our Algorithm Cmax-Fmax” with “algorithms for Pareto scheduling”.
Probably, the results of the Cmax-Fmax algorithm should be compared with the results of Pareto scheduling algorithms.

Validity of the findings

Conclusions are well stated, linked to the original research question, and limited to supporting results.

---

## Round 0.2 · accepted · Accept

Reviewers are now satisfied with the quality of the work. Congratulations

Reviewer 1 ·

Basic reporting

The authors have addressed the point of my concern. I am happy with their corrections. Hence, I would like to recommend this manuscript to be published.

Experimental design

The authors have addressed the point of my concern. I am happy with their corrections. Hence, I would like to recommend this manuscript to be published.

Validity of the findings

The authors have addressed the point of my concern. I am happy with their corrections. Hence, I would like to recommend this manuscript to be published.

Additional comments

The authors have addressed the point of my concern. I am happy with their corrections. Hence, I would like to recommend this manuscript to be published.

Reviewer 2 ·

Basic reporting

All suggestions made in the review have been implemented in the current version of the paper.

Experimental design

All suggestions made in the review have been implemented in the current version of the paper.

Validity of the findings

All suggestions made in the review have been implemented in the current version of the paper.

Additional comments

All suggestions made in the review have been implemented in the current version of the paper.